# Neural–Behavioral Relation in Phonetic Discrimination Modulated by Language Background

**DOI:** 10.3390/brainsci12040461

**Published:** 2022-03-29

**Authors:** Tian Christina Zhao

**Affiliations:** 1Institute for Learning & Brain Sciences, University of Washington, Seattle, WA 98195, USA; zhaotc@uw.edu; 2Department of Speech and Hearing Sciences, University of Washington, Seattle, WA 98195, USA

**Keywords:** linguistic experience, speech perception, mismatch response, magnetoencephalography, individual differences

## Abstract

It is a well-demonstrated phenomenon that listeners can discriminate native phonetic contrasts better than nonnative ones. Recent neuroimaging studies have started to reveal the underlying neural mechanisms. By focusing on the mismatch negativity/response (MMN/R), a widely studied index of neural sensitivity to sound change, researchers have observed larger MMNs for native contrasts than for nonnative ones in EEG, but also a more focused and efficient neural activation pattern for native contrasts in MEG. However, direct relations between behavioral discrimination and MMN/R are rarely reported. In the current study, 15 native English speakers and 15 native Spanish speakers completed both a behavioral discrimination task and a separate MEG recording to measure MMR to a VOT-based speech contrast (i.e., pre-voiced vs. voiced stop consonant), which represents a phonetic contrast native to Spanish speakers but is nonnative to English speakers. At the group level, English speakers exhibited significantly lower behavioral sensitivity (d’) to the contrast but a more expansive MMR, replicating previous studies. Across individuals, a significant relation between behavioral sensitivity and the MMR was only observed in the Spanish group. Potential differences in the mechanisms underlying behavioral discrimination for the two groups are discussed.

## 1. Introduction

It is a well-demonstrated phenomenon that listeners’ sensitivities to speech contrasts with small acoustic differences are affected by their language background. In the original series by Abramson and Lisker, discrimination of pairs of speech sounds along the Voice Onset Time (VOT) continuum was examined. First, they demonstrated that native English speakers showed a single peak in discrimination along the VOT continuum, corresponding to two phonemic categories in English (i.e., voiced vs. voiceless stops). In contrast, with the same VOT continuum, Thai speakers demonstrated two peaks in discrimination, corresponding to three categories in the Thai language (i.e., pre-voiced, voiced and voiceless) [1]. Further, when comparing English speakers to Spanish speakers, they showed that the category boundaries in perception of voice timing are significantly different between the two groups, with much shorter VOT as a boundary in Spanish speakers [2]. Such experiential effects were replicated repeatedly later on with various speech contrasts utilizing different acoustic cues [3,4,5,6].

Over the last few decades, researchers have been increasingly interested in the neural mechanisms that may underlie such linguistic effect in speech perception. The mismatch negativity (MMN), or mismatch response (MMR), is one of the most widely studied and used neural measures which has been suggested to index the neural sensitivity to sound change [7]. In its original form, a standard stimulus is repeated for the majority of the sequence (e.g., 80%) while a deviant stimulus is randomly interspersed among the standards (e.g., 20%). A difference wave is then calculated by subtracting the response to standards from the response to deviants. When measured with electroencephalography (EEG), the MMN can be observed largely in the frontocentral scalp areas, roughly 200 ms after the onset of the change in the difference wave. Both the magnitude and latency of the MMN have been suggested to show the level of sensitivity. For example, a larger change in stimulus (e.g., pure tone changing from 1000 Hz to 1032 Hz) elicited MMN with shorter latency and larger magnitude than a smaller change (e.g., pure tone changing from 1000 Hz to 1016 Hz).

In the realm of speech perception, Näätänen and colleagues were the first to demonstrate a language effect on neural sensitivities to vowel contrasts using MMN [8]. Particularly, for a vowel contrast varying on the second formant that is native to Estonians but nonnative to Finnish speakers, Estonian speakers demonstrated significantly larger MMN than Finnish speakers. This work was later followed to further examine stop consonant processing. In a series of studies focusing on VOT contrasts, researchers also demonstrated that in English speakers, a VOT contrast that crossed the phonemic boundary (i.e., /da/-/ta/) elicited larger MMN than a VOT contrast with the same acoustic distance but within the /ta/ category [9]. Further, by focusing on a VOT contrast (−10 ms vs. −50 ms) that is phonemic in Hindi but not in English, they demonstrated that, indeed, the Hindi speakers had larger MMN than the English speakers along with higher behavioral discrimination [10]. Similar effects have also been documented with many other speech stimuli [11,12].

More recently, the underlying neural generator, or the neural source, of the MMN is of research interest. Particularly, by using magnetoencephalography (MEG), which allows for robust inference of source activity, researchers have suggested that while the main contributor to the MMR is the bilateral temporal region, there is also contribution from the frontal region, likely the inferior frontal region [13,14]. So far, very limited data exist that examine the linguistic effect on MMR with a focus on the underlying sources. Only Zhang and colleagues examined this question by focusing on a pair of speech contrasts (i.e., /ra/-/la/) and measured MMR in Japanese speakers vs. English speakers using MEG [15]. Critically, the /ra/-/la/ contrast is nonnative to Japanese speakers but native to English speakers. Two types of methods were used to model the source-level activities and the results converged and demonstrated that the Japanese speakers demonstrated more widespread and longer activation for the speech contrast than the American English speakers. The authors interpreted the results to reflect a more efficient processing of native contrast in the English speakers, which is consistent with similar research using the fMRI method also using the /ra/-/la/ contrast [16]. However, the authors also cautioned an alternative account that the observed between-group difference could reflect ‘rather fundamentally different types of neural processes used by native and nonnative speakers’, instead of a difference in neural efficiency.

One way to elucidate this question is to examine the correspondence between the MMR and behavioral discrimination across individuals and compare the correspondence between groups. The rationale is as follows: if two groups of different language backgrounds rely on the same mechanisms but with different efficiency, we can expect the same MMR–behavioral discrimination correlation (i.e., the same slope) for native and nonnative speakers. Alternatively, if the two groups rely on different mechanisms, the slopes for such correlations will be different between the two groups.

However, a neural–behavioral correspondence at the individual level is rarely reported and largely assumed in the MMN/R literature. Much research has only shown parallel results between behavior and MMN/R at the group level. For example, as a group, higher behavioral discrimination was observed for a native speech contrast compared to a nonnative speech contrast. Similarly, as a group, a larger MMN/R was observed for the native speech contrast compared to the nonnative contrast. It is then often assumed that individuals who demonstrate a higher behavioral discrimination score will also have a large MMR. However, this type of analysis and subsequent results were in fact not reported [10,15]. It was therefore unclear whether the lack of neural–behavioral correspondence is due to lack of analysis/results or lack of significant findings. It is then crucial to directly examine and report the MMR–behavioral correspondence. Whether the MMR can explain a significant portion of the variance in behavioral discrimination is key to state whether MMR is indeed part of the neural underpinning of speech discrimination.

It is possible that previous studies have, indeed, conducted such correlational analyses but failed to find significant results due to lack of sensitivity in statistical methods. Indeed, in the most comprehensive review of MMN, the authors suggested that ‘in general, the MMN replicability is quite good at the group level but at the individual level, there still is ample space for further improvement before the MMN provides a reliable tool for clinics at the level of individual patients’ [7]. To that end, we employed a more exploratory machine-learning-based method to examine the correlation in addition to the traditional parametric regression analysis. The ML-based method takes data across all spatiotemporal points into consideration and, thus, may be more sensitive in detecting correlations between behavior and MMR.

The goal of the current study is thus twofold: (1) to replicate the language effect on MMR at the source level as reported by Zhang and colleagues [15], using a VOT contrast; and crucially, (2) to investigate whether individual differences in MMR at the source level can explain significant portion of variance in behavioral discrimination of speech contrast. In other words, whether there is a significant neural–behavioral correspondence across individuals, and if so, whether such neural–behavioral relation is different for native vs. nonnative speakers.

## 2. Materials and Methods

### 2.1. Participants

Monolingual English speakers (n = 15, male = 5, age = 21.4 (s.d. = 1.8)) and Native Spanish speakers (n = 15, male = 5, age = 26.0 (s.d. = 5.0)) were recruited. All participants were healthy adults with no reported speech, hearing or language disorders. All participants were right-handed (Edinburgh Handedness Quotient = 0.99 ± 0.04). All participants completed a short survey on their language and music backgrounds. The Native Spanish speakers all learned Spanish as their first language and still use the language as their predominant language. However, because they have all moved to the U.S., they also speak English to various degrees. Indeed, on the language background survey, the Native Spanish group reported higher efficiency (mean = 3.67, s.d. = 0.95) in foreign languages than the Monolingual English speakers (mean = 1. 70, s.d. = 0.67, t (28) = −6.49, *p* < 0.001). On the other hand, the Monolingual English speakers (mean = 4.67, s.d. = 4.56) reported more musical training experience (i.e., years of private lessons) than Native Spanish speakers (mean = 1.30, s.d. = 1.59, t (17.3) = 2.69, *p* =0.015, equal variance not assumed). All procedures were approved by the Institute Review Board of the University of Washington and informed consent was obtained from all participants.

### 2.2. Stimulus

Bilabial stop consonants with varying VOTs were synthesized by the Klatt synthesizer in Praat software [17]. The VOT values were −40 ms and +10 ms. The syllable with 0 ms VOT was first synthesized with a 2 ms noise burst and vowel /a/. The duration of the syllable is 90 ms. The fundamental frequency of the vowel /a/ began at 95 Hz and ended at 90 Hz. The silent gap (10 ms) and the pre-voicing (40 ms) were added after the initial noise burst to create syllables with the positive and negative VOTs. The fundamental frequency for the pre-voicing portion was 100 Hz. The waveforms of the stimulus pair, that is, voiced /ba/ (VOT = +10 ms) and the pre-voiced /,ba/ (VOT = −40 ms), are shown in Figure 1A.

Critically, this stimulus pair represents a native phonetic contrast in Spanish but not in English. Indeed, the stimulus pair was selected from a VOT continuum between −40 ms and +40 ms that was previously tested and validated [18]. Particularly, Monolingual English speakers demonstrated the category boundary to be above +10 ms while Native Spanish speakers demonstrated the category boundary to be below +10 ms. Therefore, the discrimination of the +10 ms/−40 ms (/ba/ vs. /,ba/) stimulus pair in this current study should capture the cross-linguistic difference between the two groups with different language backgrounds. That is, it represents a cross-category phonetic contrast only for the Native Spanish speakers and, therefore, should elicit higher sensitivity in them.

### 2.3. Behavioral Discrimination

An AX behavioral discrimination task was first conducted to assess the sensitivity to the speech contrast. This is the same task as used in previous studies [9,10,15,19], and a cross-linguistic effect needed be established on the behavioral discrimination of this contrast before the neural underpinning could be studied. All participants performed this task on a Dell XPS13 9333 computer running Psychophysical Toolbox [20] in MATLAB version 2016a (MathWorks. Inc., Natick, MA, USA) in a sound-attenuated booth. All sounds were delivered through Sennheiser HDA 280 Headphones at 72 dB SPL.

In an AX discrimination trial, a fixation cross was first presented for 200 ms at the center of the screen to indicate the start of the trial. Then, two speech sounds were played with a 250 ms inter-stimulus interval between them. The two speech sounds can be either the same or different (e.g., /ba/ followed by /ba/ or /,ba/). The participant was instructed to judge whether the two sounds were the same or different through key presses within 1 s. All 4 possible pairings (i.e., AA (/ba//ba/), AB (/ba//,ba/), BB (/,ba//,ba/), BA (/,ba//ba/)) were repeated 10 times in a randomized order.

The d’ values for the stimulus pair were calculated for each participant and used as the measure of sensitivity. The d’ measure takes into consideration both hit and false alarm responses, and therefore addresses the issue of response bias [21]. Specifically, the ‘hit’ is defined as when participants respond ‘different’ when sounds were different (i.e., for the AB and BA pairs), and the ‘false alarm’ is defined as when participants respond ‘different’ when the sounds were the same (i.e., for the AA and BB pairs). Then, d’ is calculated as the normalized hit rateminus the normalized false alarm rate. The hit rate for the Monolingual English speakers was 0.594 (SD = 0.318) and 0.846 (SD = 0.22) for Native Spanish speakers. The false alarm rate for the Monolingual English speakers was 0.076 (SD = 0.117) and 0.107 (SD = 0.169) for Native Spanish speakers.

### 2.4. MMR Measurement in MEG

MEG recordings were completed inside a magnetically shielded room (MSR) (IMEDCO America Ltd., IN), using a whole-scalp system with 204 planar gradiometers and 102 magnetometers (VectorView^TM^, Elekta Neuromag Oy, Helsinki, Finland). Five head-position-indicator (HPI) coils were attached to identify head positions under the MEG dewar at the beginning of each block. Three landmarks (LPA, RPA and nasion) and the HPI coils were digitized along with 100 additional points along the head surface (Isotrak data) with an electromagnetic 3D digitizer (Fastrak^®^, Polhemus, Colchester, VT, USA). In addition, a pair of electrocardiography sensors (ECG) was placed on the front and backside of the participants’ left shoulder to record cardiac activity and three pairs of electrooculogram (EOG) sensors were placed horizontal and vertical to the eyes to record saccades and blinks. All data were sampled at 1 kHz.

The sounds were delivered from a TDT RP 2.7 device (Tucker-Davis Technologies, Alachua, FL, USA), controlled by custom Python software on a HP workstation, to insert earphones. The stimulus was processed such that the RMS values were referenced to 0.01 and it was further resampled to 24,414 Hz for the TDT. The sounds were played at the intensity level of 80 dB through tubal insert phones (Model TIP-300, Natus Neurology, Pleasanton, CA, USA). A traditional oddball paradigm was used for stimulus presentation. The syllable with +10 ms VOT was used as the standard (600 trials, 80%), and the syllables with −40 ms VOT were used as deviants (150 trials, 20%) with at least two standards in between deviants. The stimulus onset asynchrony (SOA) values were jittered around 800 ms. The participants listened passively and watched silent videos during recording.

### 2.5. MEG Data Processing

All MEG data processing was carried out using the MNE-python software [22]. MEG data were first preprocessed using the Oversampled Temporal Projection (OTP) method [23] and the temporally extended Spatial Signal Separation (tSSS) method [24,25] to suppress sensor noise and magnetic interference originating from outside of the MEG dewar. Signal space projection was used to suppress the cardiac and eye movement signals in the MEG data [26]. Then, the data were low pass filtered at 50 Hz. Epochs (−100 to 900 ms) for the standards and deviants were extracted and any epochs with peak-to-peak amplitude exceeding 4 pT/cm for gradiometers or 4.0 pT/cm for magnetometers were rejected. All deviants as well as the subset of standard trials immediately preceding the deviants were averaged to calculate the evoked responses.

To estimate the location of neural generators underlying the evoked responses, each subject’s anatomical landmarks and additional scalp points were used with an iterative nearest-point algorithm to rescale the average adult template brain (fsaverage) to match the subject’s head shape. FreeSurfer was used to extract the inner skull surface (watershed algorithm) and the cortical and subcortical structures segmented from the surrogate MRI [27]. A one-layer conductor model based on the rescaled inner skull surface was constructed for forward modeling [28]. The surface source space consisted of 20,484 dipoles evenly spatially distributed along the gray/white matter boundary (i.e., ‘ico-5’). Because surrogate head models and source spaces were used for each subject, source orientations were unconstrained (free orientation). Baseline noise covariance was estimated using empty room recordings made on the same day of the MEG session. Dipolar currents were estimated from the MEG sensor data using an anatomically constrained minimum-norm linear estimation approach to obtain dSPM values at each source location [29].

The mismatch responses (MMRs) were subsequently calculated at the source level by subtracting the standards from each deviant. That is, the MMR was calculated by subtracting the vectors of standard from the vectors of deviant and then the magnitude of the vectors was calculated.

The Destrieux Atlas (i.e., ‘aparc.a2009s’ atlas in Freesurfer) was then applied to reduce the data by averaging across vertices within each label [30]. Four regions-of-interest (ROIs) were identified *a prior* based on the existing literature for further statistical analysis: left and right inferior frontal region (i.e., inferior frontal label) and superior temporal region (i.e., superior temporal label). All data reported in this study are publicly available at Open Science Framework.

### 2.6. Regression Methods

#### 2.6.1. Parametric Multiple Regression

To investigate the correspondence between the behavioral sensitivity (d’) and neural sensitivity (MMR), we first took a region-of-interest (ROI) approach to reduce the dimension of the MMR data for the parametric multiple regression analyses (see MEG data processing section for details). Four ROIs were selected *a priori* based on existing research on MMR sources, including the left and right superior temporal gyrus (STG) and inferior frontal gyrus (IFG) [13,14]. In addition, we selected the time window between 200 and 500 ms that captures maximum differences between group. The averaged MMR values were further log-transformed to reduce skewness in preparation for the multiple regression analyses.

In each multiple regression model, the main effects of the MMR value were averaged across the ROI and the selected time window and the language group (i.e., English vs. Spanish speakers) were entered. In addition, the interaction between the MMR and language group was also entered into the model to predict the behavioral sensitivity d’ (IBM SPSS Version 19.0.0).

#### 2.6.2. Machine-Learning-Based Regression

The machine-learning-based regressions were carried out using the open source scikit-learn package [31] in conjunction with the MNE-python software. This method was adopted from a previous study [32]. All spatial and temporal samples were used in this method. Specifically, for each time sample, we employed a support-vector regression (SVR) where the model uses MMR values from all 150 label regions, thus taking the spatial pattern of the MMR into consideration, to predict individual behavioral d’ value [33]. The dataset is first split into a training and a testing set (see below details regarding leave-one-out cross-validation). The MMR spatial–temporal patterns in the training set were first used to fit the model with a linear kernel function (C = 1.0, epsilon = 0.1). Once the model is trained, the MMR spatial–temporal patterns from the testing set were then used to generate predictions of the d’ value. A leave-one-out cross-validation method was used to enhance model prediction. That is, all 15 possible splits of the data (i.e., every one of the 15 participants were assigned as the testing set while the rest of the individuals (14) were the training set) were used to build 15 models and derive an averaged model. The R^2^ coefficient of determination between actual measured d’ and model predicted d’ is taken as an index of model performance. The same process was repeated for every time sample of the MMR, which generates a temporal sequence of R^2^.

To further evaluate the model performance, within each time sample, we shuffled the correspondence between the d’ and MMR spatial pattern across individuals and then conducted the same SVR analyses. In such cases, the MMR spatial pattern should bear no predictive value to d’ score and the R^2^ should reflect a model performing at chance level. We repeated this process 100 times for each time point. Then, we generated an empirical null distribution of R^2^ by pooling all the R^2^ values from each time point (i.e., 100 permutations for each time point) and we compared our originally obtained R^2^ coefficient against this distribution [34]. Specifically, we considered that if our originally obtained R^2^ value at a specific time point is larger than the 99th percentile of the empirical null distribution, then the spatial pattern of that time point can significantly predict d’ of an individual. This procedure allows a conservative way to correct for multiple comparison given the explorative nature of the analysis. A final SVR was then fit by using all the time points that were deemed significant by permutation.

## 3. Results

### 3.1. Behavioral and Neural Sensitivity to the Speech Pair Is Modulated by Language Background

To examine the effect of language background on behavioral sensitivity to the speech contrast, an independent *t* test was conducted to compare the d’ values between the two groups (IBM SPSS Statistics, version 19.0.0). Supporting our hypothesis and replicating the existing literature, the results revealed (Figure 1B) that the Native Spanish speakers exhibited significantly higher d’ than the Monolingual English speakers (t (28) = −1.83, *p* = 0.039, 1-tailed, Cohen’s d = 0.668). That is, the contrast being a phonetic contrast in one’s native language enhanced the sensitivity to the contrast.

To examine the effect of language background on neural sensitivity, as indexed by the MMR, we first visualized the MMR at the source level averaged across the whole brain for each group (Figure 1C). The first two peaks prior to 200 ms reflect the intrinsic timing difference between the standard and the deviant stimuli (100 ms vs. 130 ms in duration). The divergence between the group largely began after 200 ms and the global MMR amplitude at the source level is much more reduced in the Native Spanish speaker group.

We then further assessed the spatial distribution of MMR within each group as well as the between-group difference. The spatial distribution of the MMR over time for the Monolingual English speakers can be visualized in the left column in Figure 2 and, similarly, the spatial distribution for the Native Spanish speaker can be visualized in the middle column in Figure 2. The spatial distributions for the two groups are largely the same but with much reduced intensity in the Native Spanish group after 200 ms. Complete visualization of MMR for both groups over time can be seen in the Appendix A.

To examine the between-group difference in MMR at the whole-brain level, a spatial–temporal cluster test based on the threshold-free cluster enhancement method (TFCE) was conducted [35]. This test is nonparametric and based on permutation and is designed to allow for improved sensitivity and more interpretable output than the traditional cluster-based method. Specifically, the TFCE values were generated by summing across a series of thresholds, thus avoiding the selection of an arbitrary threshold, and then the *p* values for each spatial temporal sample were calculated through permutation. The og(p) for the between group comparison can be visualized in Figure 2 in the right column. The larger the −log(p), the smaller the p, and the more significant the between-group effect is. As can be seen, the between-group difference is significant largely after 200 ms and becomes more prominent in the right hemisphere than the left hemisphere across a wide range of regions, including the superior temporal regions (200 ms) and the inferior frontal regions (450, 650 ms) that are largely thought to underlie the MMR. Interestingly, the differences were also observed in parietal regions as well as the temporoparietal junction (TPJ) (650 ms). Complete visualization of −log(p) values over time can be seen in the Appendix A.

### 3.2. Behavioral–Neural Connection Is Affected by Language Background

#### 3.2.1. Parametric Multiple Regression

Multiple regression analyses were carried out for each ROI (see Section 2.6.1 for details on the models). For the left IFG and left STG, the models show significant fit and marginally significant fit (left IFG: R^2^ = 0.312, *p* = 0.019, left STG: R^2^ = 0.230, *p* = 0.074). Crucially, in both models, neither the MMR nor the language background were significant predictors, but the interaction between the two factors was significant and marginally significant (left IFG: B = 4.42, *p* = 0.049, left STG: B = 4.865, *p* = 0.095). As can be visualized in Figure 3A in the top row, in both ROIs, the MMR is significantly predictive of behavioral d’, but only in the Native Spanish group, not in the Monolingual English group. Conversely, similar models with right ROIs yielded nonsignificant fit (*p* > 0.1) (see Figure 3A, bottom row).

#### 3.2.2. Machine-Learning-Based Regression

In order to further explore additional spatiotemporal patterns outside of the a priori selected ones that may also be important in predicting the behavioral sensitivity, we conducted a whole-brain machine-learning-based regression using the whole MMR time series (see Section 2.6.2 for detail on the method). Given the potential differences between the two groups, we ran these analyses separately for the Monolingual English group vs. the Native Spanish group.

Using this method, we examined whether MMR could predict behavioral d’ in Native Spanish speakers and Monolingual English speakers. Consistent with the ROI analyses approach, for the Native Spanish speakers, a time window from 290 to 295 ms was deemed significant and by using that time window, the model can significantly predict individual d’ (Figure 3B, left). That is, the actual measured d’ and the model predicted d’ are significantly correlated (r = 0.48, *p* = 0.06). Critically, the areas that significantly contribute to the prediction (Figure 3B, right) show large overlap with the a priori selected ROIs. It is worth noting that the areas involve a larger frontal region including the medial frontal regions.

On the other hand, similar with the ROI analysis, the same ML-based regression did not yield any significant results for the Monolingual English Speaker group, suggesting no spatial–temporal patterns to be a good predictor of the behavioral d’ values.

## 4. Discussion

The current research extended the rich literature documenting the linguistic effect on speech processing and further examined its neural basis. Particularly, the current study examined the MMR to a speech contrast at the source level and, more importantly, the correlation between the neural MMR measure and the behavioral discrimination of the speech contrast. The speech contrast was a stop consonant contrast based on the Voice Onset Time (i.e., pre-voiced vs. voiced), which is a native phonemic contrast for Spanish speakers, but nonnative to English speakers. Monolingual English speakers and Native Spanish speakers’ behavioral discrimination of this contrast was examined along with their MMR, the most widely studied neural signature suggested to index sensitivity to sound change. The MEG-measured MMR allows a focus of examination on the source-level activities. Behaviorally, Native Spanish speakers demonstrated significantly higher sensitivity to this contrast compared to the Monolingual English speakers, demonstrating the expected linguistic effect. For the MMR at the source level, Native Spanish speakers demonstrated significantly widespread reduction compared to the Monolingual English speakers, with the difference predominantly in the right hemisphere. The behavior–MMR relation was further investigated across individuals and the results demonstrated that a significant correlation between MMR and behavioral discrimination was only observed within the Native Spanish group, but not in the Monolingual English group. Additionally, for the Native Spanish group, the cortical regions driving the behavior–MMR correlation are largely in the left frontal region.

The largely reduced MMR across multiple regions at the cortical source level for the native speakers (i.e., Native Spanish speakers), compared to the nonnative speakers (i.e., Monolingual English speakers), replicated a previous study examining linguistic effect on MMR at the source level, using a different speech contrast and populations (i.e., /ra/-/la/, Japanese vs. English speakers) [15]. This is also in line with a subsequent MEG study by Zhang and colleagues demonstrating that after intensive perceptual training to discriminate the /ra/-/la/ contrast, Japanese speakers’ MMR at the source level was also observed to be reduced [19]. These results focusing on the MMR at the source level may seem counter to the EEG-measured MMN results where MMN to native contrasts have repeatedly been shown as larger than MMN to nonnative contrasts [8,10]. However, it is important to keep in mind that while the measurement paradigms are similar for EEG and MEG, the intrinsic differences between the two technologies (i.e., measuring electric potential vs. magnetic field) dictate that they are sensitive to overlapped but different neural populations [36] and are thus picking up different signals. It is important for future research to understand more about the relationship between MEG- vs. EEG-measured MMRs and reconcile the results from these two methods to allow for unified interpretation. For example, simultaneously measured MMR using both M/EEG will allow the investigation of the relationship between MMR in EEG sensors and MMR at the source level.

The whole brain comparison of the group-level MMR between Monolingual English speakers and Native Spanish speakers revealed that the reduction is bilateral in nature, involving regions known to be important for MMR, such as the superior temporal gyrus and inferior frontal regions. Interestingly, the reduction for the Native Spanish group is much more prominent in the right hemisphere than the left hemisphere, particularly in the temporal–parietal junction (TPJ) region (Figure 2). Based on the speech processing model, spectrotemporal analysis of the speech signal at the STG level is bilateral in nature before traveling up the dorsal stream in the left hemisphere where integration of information from other modalities occurs (e.g., sensorimotor, visual) [37]. In this case, the MMR reduction in the left hemisphere and the right STG before 450 ms (Figure 2, right column) may be attributable to a more ‘efficient processing’ of the acoustic signal in the Native Spanish group. Yet, it remains unclear what the large reduction in the right hemisphere after 450 ms would entail. It was therefore crucial to examine whether any of the MMR was directly relevant for behavioral discrimination of the speech contrasts for the two groups.

As alluded to in the introduction, previous studies have hardly ever reported a correlation between MMR and behavioral discrimination. It was unclear whether it was due to lack of analysis or lack of significant findings. The current study addressed this issue directly and used two methods to evaluate whether MMR is correlated with behavioral discrimination across individuals. The results converged and demonstrated a robust correlation between behaviorally measured discrimination (d’) and MMR measured at the source level, but only in the Native Spanish group. The two types of analyses confirmed and complemented each other regarding this behavior–MMR correlation, that is, (1) traditional multiple regression analysis based on MMR extracted from a priori defined ROIs and time windows and (2) a data-driven exploratory machine-learning-based regression that takes the whole brain and the whole MMR time series into consideration. Critically, the behavioral–MMR correlation was only observed in the Native Spanish group, but not in the English speakers. This provides evidence that different mechanisms may be underlying native vs. nonnative speech MMR.

For the Native Spanish group, both types of regression analyses show that regions driving the behavior–MMR correlation are restricted to the left hemisphere and the effect seems larger in the frontal region. Further, the correlation is positive in nature, that is, the larger the MMR, the better the behavior discrimination. This result aligns well with both the theories regarding MMR as well as the speech processing model [7,14,37]. This suggests that native speakers may be processing the speech in a ‘phonetic mode’ where utilizing information from the motor planning region (e.g., left IFG) is crucial for them to distinguish two acoustically very similar speech sounds. This result is also in line with our recent studies examining the development of MMR at the source level in infants, demonstrating that the most substantial increase in MMR during the sensitive period for phonetic learning is in the left IFG region for native speech contrasts [32].

On the other hand, the lack of any behavior–MMR correlation in the Monolingual English group is surprising and puzzling, especially given the substantially enhanced MMR across the whole brain at the group level. Nonnative speech processing has generally been considered more to be at the ‘acoustic level’ of processing, such that one would expect correlation between behavior and MMR in the auditory region (e.g., STG). However, neither type of regression showed indication of such correlation. The multiple regression examined both left and right STG, while the machine-learning-based regression explored the whole brain across all time points over the MMR. One possibility is that the effect for nonnative speech is much smaller and would require a larger sample to detect the correlation. Another possibility is that attention plays a larger role in nonnative speech discrimination as attention is required for the behavioral task while MMR is measured pre-attentively. This may explain the lack of correlation between MMR and behavioral discrimination in the existing literature. Future research will need to replicate with larger samples driven by power analysis and further understand the behavioral relevance of MMR for nonnative speech. On the other hand, future research will also need to better understand the neural mechanisms for nonnative speech processing, compared to native speech processing.

## 5. Conclusions

The current study extended our current understanding of neural mechanisms underlying the linguistic effect on speech discrimination. It demonstrated that the MMR at the source level is reduced for native speakers, compared to nonnative speakers. Yet, a robust neural–behavior relation was only observed in the native speakers, suggesting potentially different mechanisms to be involved for the nonnative speakers. Future research is warranted to replicate and further elucidate the mechanisms for speech processing, particularly for the nonnative speech.

## Figures and Tables

**Figure 1 brainsci-12-00461-f001:**
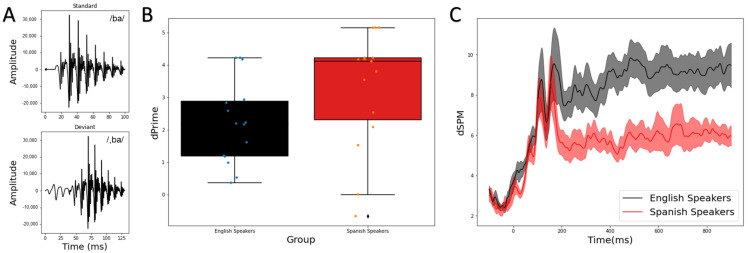
(**A**) Waveforms for the standard stimulus (**top**) and the deviant stimulus (**bottom**). (**B**) Behavioral sensitivity (d’) is different between the two groups, with Spanish speakers exhibiting higher d’ overall as this contrast is native to them. (**C**) Global MMR is different between the two groups with Spanish speakers exhibiting reduced MMR after 200 ms.

**Figure 2 brainsci-12-00461-f002:**
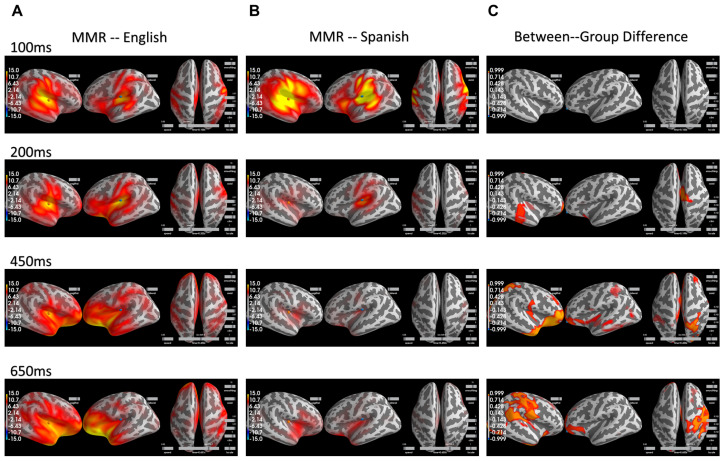
(**A**) MMR across the whole brain over time for the Monolingual English speakers. (**B**) MMR across the whole brain over time for the Native Spanish speakers. (**C**) The spatial regions that are significantly different between the two groups over time. They largely occur after 200 ms and are predominantly in the right hemisphere.

**Figure 3 brainsci-12-00461-f003:**
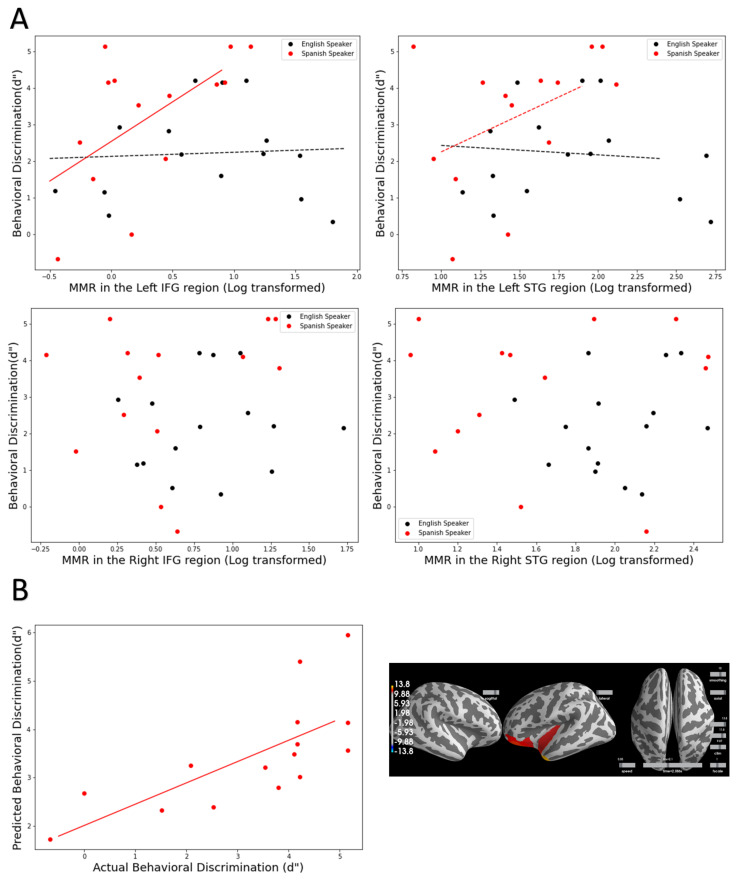
(**A**) Top row: Scatter plot between MMR in the left IFG and behavioral d’ (**left**) and between MMR in the left (STG) and d’ (**right**). The multiple regression analyses show that MMR in these two regions are significant predictors of behavioral d’ but only in the Native Spanish speakers. Bottom row: Scatter plot between MMR in the right IFG and behavioral d’ (**left**) and between MMR in the right (STG) and d’ (**right**). (**B**) A machine-learning-based regression confirmed the multiple regression based on ROIs. In the Spanish speaker group, a machine-learning-based regression can significantly predict behavioral d’ using the whole brain MMR (left column). The areas that are significant contributors to the model overlap with the a priori selected ROIs. However, no significant prediction was achieved in the Monolingual English speaker group.

## Data Availability

De-identified data and analysis code used to generate the results in this manuscript are available at Open Science Framework (https://osf.io/nyzts/?view_only=7f012b821d2a4764aa2f6a3f0ad4bd93, accessed on 6 February 2022).

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
