# Peer review of "Neural–Behavioral Relation in Phonetic Discrimination Modulated by Language Background"

_brainsci, 2022, doi:10.3390/brainsci12040461_

Round 1

Reviewer 1 Report
This study tested phonetic category discrimination in an AX task and through a mismatch negativity response measured in MEG. The phonetic contrast was native for a group of Spanish listeners but non-contrastive for the group of English listeners. The MMR at the source level was reduced for the native speakers compared to the nonnative speakers, as expected. Only for the native speakers, a relationship between behavioral and neural data was found.
I have a few questions and concerns:
One important issue is that the interpretation of the MNN needs to be discussed. The MMN is described as indexing “neural sensitivities”. However, the neural mechanism underlying the MMN are not yet well understood. I believe there are even studies questioning that it reflects perceptual processes. Either way, the interpretation of the results of this study here critically hinges on understanding what the MNN (or in this case the MMR) even measures.
I have a related concern about the AX task. I would like to read about the reasons why this task was chosen over others. Different discrimination tasks seem to reflect different processes, see Gerrits & Schouten, 2004 for an overview and for specific concerns about the AX task. Gerrits (2001), for example, found differences in performance across AX vs AXB tasks. The question therefore here is what is it that is actually measured by the AX task. Understanding the underpinnings of the behavioral task is crucial for this study.
Combining these two concerns, what does any relationship between behavioral and neural data actually reflect, given questions about what the MMR and the AX task actually measure. This needs to be discussed and explained. (I also think that these choices should be justified already from the beginning of the paper, so this is not simply fixed by adding discussion to the end).
I found it odd that participants were not also tested on a full continuum between bilabial stop consonants to show that a) the Spanish speakers accept these sounds as realistic and b) to show that indeed a phonetic category boundary can be found for these synthetic stimuli for the Spanish speakers (and that therefore the selected two stimuli span this boundary). It would also show how perceptually close or different a stimulus with -40ms VOT and one with +10 VOT are for those listener groups. I think that this data is crucial in interpreting the results presented here.
I am wondering that in particular as (looking at Figure 3) it seems that there is substantial overlap of English and Spanish listeners in their behavioral performance and that the steepness of the regression line for Spanish speakers (red line) is perhaps mainly due to two Spanish speakers with a very low d’, which might be a problem given that the behavioral-neural correspondence was only found for that group. I think it is typically the case to expect participants to exceed at certain threshold of d’ to be included (e.g. we have used dprime <.3 before). Please also report the hit rate and false alarm rates.
Some more issues:
Please define “Native Spanish speakers”. More importantly, at the end of the Introduction, please explain why Spanish speakers were chosen and give more details here on how VOT contrasts differ in Spanish vs. English. It would also help to say already in the introduction that the test “language” is Spanish (i.e., synthesized Spanish).
The stimuli were synthesized but no information is provided on how the values used for synthesis were chosen and what justifies their selection. For example, it is unclear why VOT values of -40ms and +10ms were chosen (and not e.g. -100ms and 0 VOT). This needs to be related back to what typical values are in Spanish for these two categories and pilot data that lead to these choices. Also add an IPA transcription of these two consonants (also to Figure 1A).
Information about power (to justify sample sizes) and effect sizes should be provided.
p.2 First paragraph: I think it would be easier to follow if first talked about how MMN picks up the pattern of categorical perception within English listeners, before talking about how it is modulated by language experience.
p.2 2nd paragraph: I think expanding the description of Zhang et al here would be beneficial. I don’t know what “smaller ECD cluster and shorter ECD duration” means and why there are multiple ways to interpret the results. Given that these multiple possible explanations are used to motivate the current study, this needs to be clearer.
p.2 3rd paragraph: This is odd – because either the current study looks at neural-behavioral correspondences at the group level. Then the question is why so much space is dedicated to saying that this cannot be done yet for the individual level. Or, the current study looks at these correspondences at the individual level, but then why is that justified here, given that we learn here that this cannot be done yet?

Author Response

This study tested phonetic category discrimination in an AX task and through a mismatch negativity response measured in MEG. The phonetic contrast was native for a group of Spanish listeners but non-contrastive for the group of English listeners. The MMR at the source level was reduced for the native speakers compared to the nonnative speakers, as expected. Only for the native speakers, a relationship between behavioral and neural data was found.
I have a few questions and concerns:
One important issue is that the interpretation of the MNN needs to be discussed. The MMN is described as indexing “neural sensitivities”. However, the neural mechanism underlying the MMN are not yet well understood. I believe there are even studies questioning that it reflects perceptual processes. Either way, the interpretation of the results of this study here critically hinges on understanding what the MNN (or in this case the MMR) even measures.

Thank you for raising this question. We have added in additional information in introduction and hopefully have made our viewpoint clearer. Indeed, we are trying to understand a bit more of what MMR is actually measuring. Previous research largely relies on group level analysis (e.g., for a larger change in pitch, behaviorally people detect the change better and they also demonstrate larger MMR). However, a behavior-MMR correlation across individuals have been largely assumed and not reported. Establishing this correlation would be a key to suggest that MMR is indeed a neural underpinning of behavioral discrimination. In the current study, we are testing this question directly, that is, whether a behavior-MMR correlation exists and further, whether it is modulated by language backgrounds.

I have a related concern about the AX task. I would like to read about the reasons why this task was chosen over others. Different discrimination tasks seem to reflect different processes, see Gerrits & Schouten, 2004 for an overview and for specific concerns about the AX task. Gerrits (2001), for example, found differences in performance across AX vs AXB tasks. The question therefore here is what is it that is actually measured by the AX task. Understanding the underpinnings of the behavioral task is crucial for this study.

Thank you for raising this interesting question. Our rationale for selecting the AX task was purely for replication purposes, given the recent studies using M/EEG used the same behavioral task to confirm that there indeed was a behavioral difference for the stimulus pair, before moving on to neural measures (e.g., Shama & Dorman, 1999, 2000, Zhang et al, 2005, Zhang et al., 2009). We have added in this information in section 2.3.
We agree that the particular behavioral task may have a significant effect on categorical perception, but that effect is more pronounced in vowel perception. Indeed, research has proposed other perception model for vowels (e.g., perceptual magnetic effect). However, for stop consonants, the categorical nature of the perception has been shown fairly robust across different types of tasks. Given our results, along with the previous studies all report robust cross-linguistic effects, we believe that the AX discrimination can reflect the sensitivity for a stop consonant contrast and can capture effects from having different language background. Building on this, the current study, along with previous ones, aimed to examine the neural underpinning of this cross-linguistic difference in sensitivity.

Combining these two concerns, what does any relationship between behavioral and neural data actually reflect, given questions about what the MMR and the AX task actually measure. This needs to be discussed and explained. (I also think that these choices should be justified already from the beginning of the paper, so this is not simply fixed by adding discussion to the end).

Taken together our revision in both introduction and discussion, we hope we have clarified our viewpoint on this question.  

I found it odd that participants were not also tested on a full continuum between bilabial stop consonants to show that a) the Spanish speakers accept these sounds as realistic and b) to show that indeed a phonetic category boundary can be found for these synthetic stimuli for the Spanish speakers (and that therefore the selected two stimuli span this boundary). It would also show how perceptually close or different a stimulus with -40ms VOT and one with +10 VOT are for those listener groups. I think that this data is crucial in interpreting the results presented here.

Thank you for this concern. We have added in the 2.2 stimulus section information regarding the stimulus pair. Indeed, the behavioral data regarding category boundaries on these stimuli were validated and published before (Zhao & Kuhl, 2018). We added description of the results and rationale for selecting this pair based on the previous results. To summarize, English and Spanish speakers both demonstrated clear category boundaries on this VOT continuum with different category boundaries either above or below +10ms. Therefore, we believe discrimination between the +10ms/-40ms pair should nicely capture the cross-linguistic difference and selected this pair in our effort to further understanding the neural underpinning. We conducted the additional AX discrimination with our stimulus pair and reported the results in this paper and demonstrated that behaviorally, this pair indeed differentiated the two groups, replicating previous results.

I am wondering that in particular as (looking at Figure 3) it seems that there is substantial overlap of English and Spanish listeners in their behavioral performance and that the steepness of the regression line for Spanish speakers (red line) is perhaps mainly due to two Spanish speakers with a very low d’, which might be a problem given that the behavioral-neural correspondence was only found for that group. I think it is typically the case to expect participants to exceed at certain threshold of d’ to be included (e.g. we have used dprime <.3 before). Please also report the hit rate and false alarm rates.

Thank you for raising this question. Our rule for outlier exclusion is if it’s 3 SD outside of the mean which none of the data points are. A large overlap in individual data is generally expected given the individual differences within the groups. We have also repeated our regression analysis while excluding the two Spanish speakers with low d’ as the reviewer suggested, the results actually became more significant (p for the interaction for the left IFG went from 0.049 to 0.003). We therefore kept our results as it is.
We have also added in more information regarding the d’ as well as hit and fa rate in section 2.3.

Some more issues:

Please define “Native Spanish speakers”. More importantly, at the end of the Introduction, please explain why Spanish speakers were chosen and give more details here on how VOT contrasts differ in Spanish vs. English. It would also help to say already in the introduction that the test “language” is Spanish (i.e., synthesized Spanish).

Thank you for raising this issue. We have added in information in section 2.1 and 2.2 to clarify these points.

The stimuli were synthesized but no information is provided on how the values used for synthesis were chosen and what justifies their selection. For example, it is unclear why VOT values of -40ms and +10ms were chosen (and not e.g. -100ms and 0 VOT). This needs to be related back to what typical values are in Spanish for these two categories and pilot data that lead to these choices. Also add an IPA transcription of these two consonants (also to Figure 1A).

Thank you for this question. We have added in information regarding the stimulus in section 2.2 and the IPA symbols.

Information about power (to justify sample sizes) and effect sizes should be provided.

Thank you for raising this issue. We have added the effect size for the t test while R2 has already been provided for the multiple regressions. We did not conduct a power analysis since our sample sizes were larger than previous studies. However, we do agree that this may be an issue for the individual differences and we have add in the discussion section about this issue.

p.2 First paragraph: I think it would be easier to follow if first talked about how MMN picks up the pattern of categorical perception within English listeners, before talking about how it is modulated by language experience.

We have revised introduction significantly and the flow should be clearer now.

p.2 2nd paragraph: I think expanding the description of Zhang et al here would be beneficial. I don’t know what “smaller ECD cluster and shorter ECD duration” means and why there are multiple ways to interpret the results. Given that these multiple possible explanations are used to motivate the current study, this needs to be clearer.

Thank you for this suggestion. We have revised the description of the results and hopefully it’s clearer now.

p.2 3rd paragraph: This is odd – because either the current study looks at neural-behavioral correspondences at the group level. Then the question is why so much space is dedicated to saying that this cannot be done yet for the individual level. Or, the current study looks at these correspondences at the individual level, but then why is that justified here, given that we learn here that this cannot be done yet?

Thank you for raising this point. We hope our revised version clarifies our viewpoint better. Previous studies argue that MMR is indicative of sensitivity based on group-level analysis. For example, as a group, listeners demonstrate a larger MMR to a larger change in acoustics (e.g., 100 Hz difference in pure tone, compared to a smaller change (e.g., 30 Hz). However, correlation at the individual level is largely not examined, assumed, or not observed in previous studies. That is, to say, an individual who has larger MMR should have better discrimination.

In previous studies on speech perception, cross-linguistic effects have been observed at the group level (e.g., MMR to native contrast is larger than nonnative contrast). However, whether this is true across individuals (i.e., someone who discriminate better has larger MMR) is largely unknown. Previous studies have suggested that MMR might not sensitive enough to detect such neural-behavioral correspondence at the individual level. However, it is just simply not shown in previous studies whether such correlational analyses were done and what the results were.

So it might be that they all did the analyses, and they weren’t significant, leading to the suggestion that MMR might not be good enough for individual differences. Or it could be that such analyses were simply not done. For the first possibility, it is also possible that a lack of sensitive statistical method led to a lack of significant correlation. Therefore, in this study, we highlighted the correlational analyses to address the question on individual differences, and we further deployed a more sensitive ML-based correlational analysis in addition to the traditional parametric regression analysis.

Reviewer 2 Report

This manuscript investigated the differences in native English language perception between native vs non-native English speakers via MMR 'feature' at the source level with MEG data. This has already been clearly shown however, reproducing the work for a different language group (Spanish) has its own merit. The other objective of this manuscript, i.e., individual level differences in MMR at the source level can explain the neural-behavioral relation differences across different groups of speakers, has not been clearly shown. Perhaps, the description is not clear that shows individual level analyses. Currently, it seems that the regression analysis was done for each group separately for different ROIs and a difference was observed. I would advise the author to clearly explain their analysis for this section.

General Comments: Please describe your training and testing strategy, i.e., number of samples per training/test. I see 14 d' values in the figure. They were test samples? The author previously described 4 pairs (AA,AB, BB, BA) each with 10 repetitions. Has the analysis been done for each pair. The figures correspond to which type of contrast. The leave one cross validation is by leaving one individual's test data or all data. These details are missing. With the limited dataset, is it even possible to do individual subject regression? If not, I would advise the author to change the objective in line to the actual analysis conducted. Please take all the methodological details from results section and give separate subheadings in the method section.

Author Response

This manuscript investigated the differences in native English language perception between native vs non-native English speakers via MMR 'feature' at the source level with MEG data. This has already been clearly shown however, reproducing the work for a different language group (Spanish) has its own merit. The other objective of this manuscript, i.e., individual level differences in MMR at the source level can explain the neural-behavioral relation differences across different groups of speakers, has not been clearly shown. 
Perhaps, the description is not clear that shows individual level analyses. Currently, it seems that the regression analysis was done for each group separately for different ROIs and a difference was observed. I would advise the author to clearly explain their analysis for this section.

We have revised our 3.2 section to clarify our multiple regression models. The two groups, and the interaction related to the group were entered into the same model. We tested whether the interaction term was significant. If the interaction term was significant, that showed that the slopes for the two groups were different.

General Comments: Please describe your training and testing strategy, i.e., number of samples per training/test. I see 14 d' values in the figure. They were test samples? The author previously described 4 pairs (AA,AB, BB, BA) each with 10 repetitions. Has the analysis been done for each pair. The figures correspond to which type of contrast.

Thank you for this question. We have added in additional information regarding the d’ calculation and the hit and false alarm. All 40 trials were used to calculate this data.

The leave one cross validation is by leaving one individual's test data or all data. These details are missing. With the limited dataset, is it even possible to do individual subject regression? If not, I would advise the author to change the objective in line to the actual analysis conducted. Please take all the methodological details from results section and give separate subheadings in the method section.

Thank you for this question. The leave-one-out-cross-validation is a method to use when the N is small. That is, in one round, one individual’s is designated as the test data while the rest of the group (n=14) serve as the training set. The model is trained on the training set and then the model is given the test set’s MMR to generate a ‘predicted d’. This process was then repeated for all individuals (15 times) and outcome model is the average of all models, and therefore improve the model accuracy.  Critically, because we are using all spatial temporal data points of the MMR for each individual (~14000 spatial points x ~600 temporal points x 14 individuals). The model was actually given a lot of data for training and the model outcome should be robust.

We have also reorganized the section with subheadings and moved the method details to the method section.

Round 2

Reviewer 2 Report

Please see my comments below:
Objective 2 still says individual differences in MMRwhich is not clear. what does 'individual' refer to here?
Please type down the exact stimuli used.
Describe what is an AX behavioral task. what are the All 4 possible pairings (i.e., AA, AB, BB, BA. speech sounds? What are the two speech sounds?
Please add more details about parametric multiple regression model in the method section. A lot of methodological details are in the result section now for this method.
For machine learning model you explained that you performed leave-one out cross validation strategy. But you also mentioned in your manuscript that data were divided randomly between train and test. Which one is true. Kindly rectify.
Please show scatter plots for all 4 ROIs for parametric regression.
Please show correlation between actual d' and predicted d' for both group of speakers.
Regression analysis figures are blurry. Kindly improve the resolution.

Author Response

Objective 2 still says individual differences in MMRwhich is not clear. what does 'individual' refer to here?

‘Individual differences’ is a widely accepted term in psychological research that described the intrinsic differences or variability within a group (https://dictionary.apa.org/individual-differences). Here, what the aim 2 targets is whether within these groups, the individual with larger MMR are indeed the ones who also scored higher (higher d’) on the behavioral task. This idea was discussed in depth in introduction paragraph 5-7. We have also further edited the sentence to make it clearer.

Please type down the exact stimuli used.

The stimulus information is described in section 2.2. We further edited it to make it clear that the stimulus pair included a voiced bilabial stop consonant (/ba/ with VOT of 10ms) and a pre-voiced bilabial stop consonant (/,ba/ (VOT = -40ms).

Describe what is an AX behavioral task. what are the All 4 possible pairings (i.e., AA, AB, BB, BA. speech sounds? What are the two speech sounds?

We have further edited the 2.3 to make it clear how the task works.

Please add more details about parametric multiple regression model in the method section. A lot of methodological details are in the result section now for this method.

We have moved the data preparation section for multiple regression to under 2.6.1 section.

For machine learning model you explained that you performed leave-one out cross validation strategy. But you also mentioned in your manuscript that data were divided randomly between train and test. Which one is true. Kindly rectify.

Thank you for catching this error. We have removed the word ‘randomly’ and added further description of the leave-one-out cross validation method.

Please show scatter plots for all 4 ROIs for parametric regression.

We have added the scatter plots for the other 2 ROIs in Fig. 3A.

Please show correlation between actual d' and predicted d' for both group of speakers.

The English group cannot be plotted because no spatiotemporal patterns were deemed good (significant) predictor of d’ through the permutation process, therefore, there was no good model to generate predicted d’ for the English group.

Regression analysis figures are blurry. Kindly improve the resolution.

We have improved the font of the figures. Hopefully it has addressed the issue.